# Analysis of the Determinants of Agriculture Performance at the European Union Level

Oana Coca, Diana Creangă, Ștefan Viziteu *, Ioan Sebastian Brumă and Gavril Ștefan

Faculty of Agriculture, "Ion Ionescu de la Brad" Iasi University of Life Sciences, 700490 Iasi, Romania
* Correspondence: stefan.viziteu@yahoo.com

**Abstract:** In the context of the increase in energy prices and, as a consequence, on other inputs on the global and European market, the study aims to analyze the performance of the European Union's agriculture through the lens of the correlations and links that are established between different determinants/factors and which provide a much clearer picture of the agricultural sector than the analysis of the result/output and its contribution to GDP. The working methodology consisted in the comparative analysis of the statistical data related to the Member States and the aggregated general data for the EU 27 using the EUROSTAT data by processing them in order to achieve the proposed goal. The results indicate a high level of performance for a number of states that are not considered very important in this area and, in contrast, a lower level of performance for a number of states considered at the top of the pyramid in terms of performance. The derived indicators used and calculated within the research can represent basic elements in the fundamental analysis of the agricultural activity performance of a country and the European Union as a whole.

**Keywords:** agriculture performance; European Union farms; output/input ratio; agricultural output; arable land

## 1. Introduction

Agriculture has a global and fundamental role in human life, representing the basic sector in ensuring food for the population, generating employment and rural development, promoting conservation of natural resources and contributing to the development of other sectors of the world economic and social activity [1–3]. Being a strategic element of the economy and ensuring food security [4], agriculture have several specificities [5], each country developing a specific structure [6] but with similarities to specific groups of countries depending on different performance indicators.

Although it is not a major contributor anymore to the national economic growth and its direct impact on the macroeconomic level is decreasing at the same time with the socio-economic development of countries [7–9], agriculture remains a *sine qua non* domain for the world and humanity.

The performance of agriculture is a result of several determinants and particular factors with relationships of conditioning, complementarity, or competitiveness established between them. The Common Agricultural Policy (CAP) reforms and the enlargement of the European Union brought changes in the performance of the agricultural sector for the member countries [10], farms' economic performance being a result of structural, process and behavioral factors [11].

The main purpose of the paper is to analyze the performance of agriculture in the European Union by identifying its main determinants and the correlations between the factors that ensure the elements of performance. The current paper aims at generating cross-country comparisons and providing new insights into EU agricultural performance in terms of land use, production, labor, or technical and economic efficiency.

The research focused on the comparative analysis between the member countries of the European Union, not only on being an analysis of the EU agriculture as a whole,

the differences/gaps highlighted, the correlations established, and the new indicators calculated highlighting the performance from a different point of view and through a more accurate set of factors. The level of performance is sometimes reflected by indicators other than those referring only to results (e.g., agricultural output, gross value added (GVA), factor income), such as derived indicators (e.g., output–input ratio) or GFCF (gross fixed capital formation)/UAA (utilized agricultural area).

The main questions answered by the current research are:

1. What are the determinants of agricultural performance in the EU?
2. What are the relationships between the determinants of EU agricultural performance and how are Member States ranked according to them?
3. How do EU countries differ and group according to the analyzed agricultural influencing factors?

The novelty of the article is given by the selected indicators and the new connections established in order to ensure a relevant assessment of the performance, targeting a segment of interest less addressed by research papers, and looking at the factors involved through the prism of the conjugate influence they have on the overall results. This paper is an attempt at filling a gap in the research regarding similarities in EU countries not in terms of the results but regarding a group of selected indicators that highlight the current situation and create the basis for forecasts of the evolution of the agricultural sector at the macroeconomic level.

## 2. Literature Review

Investigating specialized literature is one of the basic steps in scientific research. This activity allows the substantiation of the documentation process and the application of specific methods and techniques, but also the identification of limits, the establishment of opportunities to expand research, and a different approach by changing the parameters in order to find new solutions and elements of added value.

The general topic mentioned, related to the agricultural sector and its performance has also been addressed in multiple papers that addressed different specific aspects adapted to the scientific approach and to the research objectives.

Reiff [10] analyzed differences in the agricultural performance of the EU countries for the time span of 2010-2013. The data source was World Bank indicators and using Ward's clustering method they grouped countries into six classes indicating a significant disparity in the performance of agriculture between the old and new Member States.

Another study [12] identifies the differences in land, capital, and labor endowments for different agricultural production types in the EU 28 at the level of year 2015, using Ward Cluster Analysis for grouping EU regions but also DEA model and ANOVA analysis to assess the significance of differences in the technical efficiency of agricultural holdings for each country.

Agricultural sustainability and production factors (land, labor, capital, and entrepreneurship) are analyzed by Magrini [13] in a study regarding 26 countries of the EU in the period 2004–2018 (15 years) using computed Eurostat data showing distinctive trends of CAP subsidies in three groups with different tendencies.

The most important determinants of competitivity in the agricultural sector are studied in another research [14] which identifies differences in the agricultural potential between the EU and the USA and indicates which groups of countries generate competitivity in this sector using data available for the year of 2016. The study also uses cluster analysis with the Euclidean distance in order to separate competitive countries regarding agricultural performance and points to Germany, the Netherlands, France, Denmark, and Belgium as countries that rise at the same competitive standard as the USA level of agricultural development.

Zsarnóczai and Zéman [15] analyze the correlations among different economies of EU-12 Member States, comparing different agricultural performance determinants (output value of the agricultural industry, input productivity, agricultural GVA, subsidies, labor, and agricultural income per AWU) by using the SPSS program and other statistical methods,



generating a dendrogram based on Ward Cluster Analysis. The study focused on the 2010–2016 period and stated that EU-12 countries achieved a higher increase of output and agricultural GVA than EU-28 countries.

Many studies address, in addition and directly related to the research topic addresses more complex issues regarding environmental performances agriculture [16–20] of sustainability [21–26] or food security and food policies [27–30] because agricultural performance cannot be described without the general framework of which it is a part. Agricultural policies became increasingly complex [31] and this is why the CAP (Common Agricultural Policy), as the root of the main support measures taken in agriculture, is analyzed by different research papers [12,32,33].

The topic of determinants and factors creating real agriculture performance was approached by different authors [34–39] who quantified the influence of each indicator in the final results. Some studies even focused on one country within the EU [40–43], on different regions of the EU [44,45] or on a comparison between one country and EU [46,47], all of which can be used in achieving the current research goals by extrapolation.

The approach to measurement profitability in agriculture is still evolving [48], the policy instruments such as subsidies having an ever-increasing influence by being a strategic tool [12,49,50], competitiveness and profitability generating economic growth and increasing the income for agricultural holdings and the wellbeing of farmers [51], EU agriculture being variable in terms of resources and relationships between production factors [52].

The above literature review is the base of our scientific approach and highlights the necessity of a new approach to European agricultural performance, from a different perspective and angle, using derived indicators and more current data in order to respond to the new challenges that this field of interest faces.

## 3. Materials and Methods

### 3.1. Data and Hypotheses

The main source of data used and processed to meet the proposed purpose was represented by Eurostat statistics regarding Agriculture, forestry and fisheries [53] which provided essential raw material at the EU-27 level up to and including the year 2020. Adjacent sources of information have been the European Commission, Directorate-General (DG) Agriculture [54], FAO (Food and Agriculture Organization of the United Nations) [55] or OECD (The Organization for Economic Cooperation and Development) [56].

In order to determine the performance of agriculture at the level of the European Union and each Member State, a series of terms specific to the agricultural sector and the agrarian economy was used, which require a brief definition in order to integrate them into the comparison groups and in the calculations performed.

Utilized agricultural area (UAA) represents the total area taken up by arable land, permanent grassland, permanent crops, and kitchen gardens used by the holding, regardless of the type of tenure or whether it is used as a part of common land [57].

An agricultural holding, holding or farm, is a single unit, both technically and economically, operating under a single management and which undertakes economic activities in agriculture within the economic territory of the European Union, either as its primary or secondary activity [57].

The standard output (SO) of an agricultural product (crop or livestock), is the average monetary value of the agricultural output at farm-gate price, in euro per hectare or per head of livestock. The sum of all the SO per hectare of crop and per head of livestock in a farm is a measure of its overall economic size, expressed in euro.

Final agricultural output measures the value of agricultural products which, free of intra-branch consumption, is produced during the accounting period and, before processing, is available for export and/or consumption [56].

Agricultural factor income measures the remuneration of all factors of production (land, capital, labor) regardless of whether they are owned or borrowed/rented and represents all the value generated by a unit engaged in an agricultural production activity.

Agricultural factor income (net value added at factor costs) = Value of agricultural production − variable input costs (fertilizers, pesticides, feed, etc.,) − depreciation − total taxes (on products and production) + total subsidies (on products and production) [56].

Gross fixed capital formation (GFCF), consists of resident producers' investments, deducting disposals, in fixed assets during a given period. It also includes certain additions to the value of non-produced assets realized by producers or institutional units. Fixed assets may be tangible or intangible assets produced as outputs from production processes that are used for more than one year [58].

Annual work units (AWUs) are defined as full-time equivalent employment (corresponding to the number of full-time equivalent jobs), which are calculated by dividing total hours worked by the average annual number of hours worked in full-time jobs within the economic territory, being a measure of labor productivity in agriculture [21].

Related to the paper's objective, the established working hypotheses are as follows:

**Hypothesis 1.** *Gross fixed capital formation has a direct positive influence on total agricultural output ensuring EU agricultural performance.*

**Hypothesis 2.** *The output–input ratio for the EU Member States as performance indicator is related to agricultural output and correlates with agricultural results for each country.*

**Hypothesis 3.** *The EU agriculture development among the EU Member States register a common foundation/base as concentration degree.*

*3.2. Methodology*

The most appropriate methods for the scientific approach were chosen to illustrate the results and correspondences between different factors generating performance in agriculture as faithfully as possible.

In order to highlight the gaps between countries' agricultural development, a comparative analysis was used, describing and explaining the similarities and differences of situations or consequences. It can also be applied among a large scale of social units such as regions, nations, societies, and cultures [59]. For a clearer visualization, tabular or graphical comparisons of simple or derived indicators were used.

Input–output analysis allowed an examination of the contribution of the primary sector to the general economy and the impact of the CAP on local agricultural development [60]. Outputs can be expressed in terms of physical quantity (vegetal or animal yields), as well as in terms of value (turnover, value added, profit); and inputs can be represented by various production factors, such as property assets, technical capital, natural factors, labor quality [38].

The study regarding correlation between different factors contributing to agricultural performance was based on Pearson's Correlation, which identifies linear correlation between variables $X$ and $Y$, Pearson's correlation coefficient $r$ ($X$, $Y$), returns values between −1 and 1. If $r = 1$, then there is a positive linear correlation between variables $X$ and $Y$, i.e., if $r = -1$, then there is a negative correlation between variables $X$ and $Y$. In the event of linear independence, the correlation coefficient is equal to zero ($r = 0$), and values of variables $X$ and $Y$ are scattered independently of one another [42,61].

Cluster analysis using Ward's method is based on a classical sum-of-squares criterion, producing significant groups [62]. The clustering process is applied to highlight the grouping of countries in relation to the level reached by different indicators. At the first stage, in EU-27, we isolated areas (aggregations of EU-27 countries) significantly different in terms of output/input ratio, gross fixed capital formation per UAA and factor income in 2020. For that purpose, we carried out a cluster analysis using Ward's method with the use of Euclidean distance, including 27 countries' data, using Eurostat statistics. The results of the cluster analysis made it possible for us to isolate the EU region aggregations characterized by similar factors.

The methodology related to the current research is limited to the analysis of economic indicators based on Eurostat data, the use of specific elements to establish correlations between different determinants for the performance of the agricultural sector and grouping countries according to the similarity generated by three associated performance criteria.

The methods used within the scientific research have a validity proven by other studies/similar sources investigated, but their use together in order to express the performance of the agricultural sector at the level of the European Union, the selection of indicators presented in the current paper studied through comparative analysis with the other Member States and with the level EU 27 average, as well as the calculation of the derived indicator (output/input ratio) and its influence on generating similarity groups, are original elements of the research that differentiate the current approach from those existing in other studies in the field.

The approach of the topic followed the path appropriate to the scientific nature of the paper starting from the general (analysis of agricultural land and a utilized agricultural area) to the particular by analyzing the structural elements of the agricultural sector in the most detailed way possible and identifying those determinants with major influence in the efficiency of agricultural activity for each country.

The structure of the research aimed at the year 2020 without making an analysis of the results dynamics or a comparison with the previous levels of the performance indicators and the associated determinants (topics that may represent the subject of future research), being aware that the correlation between variables/determinants indicate the connection between them and does not necessarily imply causality between these. The methodology used can be extended or reconfigured for different scenarios in order to frame the agricultural performance.

## 4. Results and Discussion

An overview on EU agriculture is offered by identifying the main production factor that constitutes the support of agricultural activity, namely the land [63] mainly utilized agricultural area (UAA) as indicator [60] land being a fixed factor [64]. The processed data from Eurostat statistical resources indicate for the year 2020 a total value of utilized agricultural area (UAA) of 157,416 million hectares out of a total of 190,131 million hectares of farm area. Regarding arable land, the largest share is held by France (17.4%), followed by Germany (11.9%), Spain (11.9%), and Poland (11.4%) (Table 1).

**Table 1.** EU agricultural land use, 2020 (thousand hectares).

| No. | Country/Item | Farm Area | | UAA | | Arable Land | |
|-----|-------------|-------|---------|-------|---------|-------|---------|
| | | Value | % of EU | Value | % of EU | Value | % of EU |
| 1 | Austria | 4798 | 2.5 | 2603 | 1.7 | 13,229 | 1.3 |
| 2 | Belgium | 1392 | 0.7 | 1368 | 0.9 | 8693 | 0.9 |
| 3 | Bulgaria | 4907 | 2.6 | 4564 | 2.9 | 33,184 | 3.4 |
| 4 | Croatia | 1647 | 0.9 | 1505 | 1.0 | 8880 | 0.9 |
| 5 | Cyprus | 146 | 0.1 | 134 | 0.1 | 1022 | 0.1 |
| 6 | Czechia | 4923 | 2.6 | 3493 | 2.2 | 24,767 | 2.5 |
| 7 | Denmark | 3152 | 1.7 | 2630 | 1.7 | 23,734 | 2.4 |
| 8 | Estonia | 1211 | 0.6 | 975 | 0.6 | 6929 | 0.7 |
| 9 | Finland | 5305 | 2.8 | 2282 | 1.4 | 22,557 | 2.3 |
| 10 | France | 29,239 | 15.4 | 27,365 | 17.4 | 170,394 | 17.4 |
| 11 | Germany | 18,314 | 9.6 | 16,595 | 10.5 | 116,638 | 11.9 |
| 12 | Greece | 4066 | 2.1 | 3917 | 2.5 | 15,021 | 1.5 |
| 13 | Hungary | 6439 | 3.4 | 4922 | 3.1 | 40,280 | 4.1 |
| 14 | Ireland | 5215 | 2.7 | 4920 | 3.1 | 12,098 | 1.2 |
| 15 | Italy | 16,462 | 8.7 | 12,524 | 8.0 | 71,977 | 7.3 |
| 16 | Latvia | 2830 | 1.5 | 1969 | 1.3 | 13,333 | 1.4 |

**Table 1.** *Cont.*

| No. | Country/Item | Farm Area | | UAA | | Arable Land | |
|---|---|---|---|---|---|---|---|
| | | Value | % of EU | Value | % of EU | Value | % of EU |
| 17 | Lithuania | 3086 | 1.6 | 2915 | 1.9 | 22,373 | 2.3 |
| 18 | Luxembourg | 138 | 0.1 | 132 | 0.1 | 623 | 0.1 |
| 19 | Malta | 11 | 0.0 | 10 | 0.0 | 78 | 0.0 |
| 20 | Netherlands | 1947 | 1.0 | 1818 | 1.2 | 10,082 | 1.0 |
| 21 | Poland | 16,662 | 8.8 | 14,784 | 9.4 | 111,472 | 11.4 |
| 22 | Portugal | 5121 | 2.7 | 3964 | 2.5 | 10,367 | 1.1 |
| 23 | Romania | 13,787 | 7.3 | 12,763 | 8.1 | 85,707 | 8.7 |
| 24 | Slovakia | 3016 | 1.6 | 1863 | 1.2 | 13,253 | 1.4 |
| 25 | Slovenia | 907 | 0.5 | 483 | 0.3 | 1731 | 0.2 |
| 26 | Spain | 28,930 | 15.2 | 23,914 | 15.2 | 117,147 | 11.9 |
| 27 | Sweden | 6478 | 3.4 | 3006 | 1.9 | 25,382 | 2.6 |
| 28 | EU 27 (from 2020) | 190,130 | 100.0 | 157,416 | 100.0 | 980,948 | 100.0 |

Source: Authors' calculation, according to Eurostat (2020) data.

The map of EU UAA indicates that alongside the countries mentioned before, an important place is also held by Romania with 8.11% of the total and by Italy with 7.3%. (Figure 1).

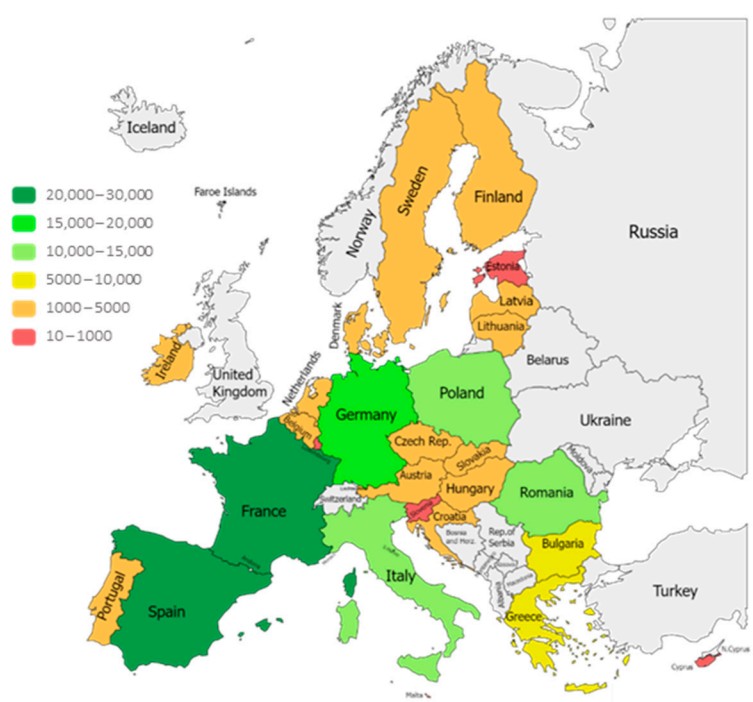

**Figure 1.** Map of EU UAA 2020 distribution (thousand hectares). Source: Authors' calculation and representation, according to Eurostat (2020) data.

For most of the EU countries, the share of arable land in UAA is predominant, but there are also a few countries (Ireland, Slovenia, Portugal) where the highest percentage is owned by permanent grassland while kitchen gardens- outdoor area is significant only for Malta. Permanent crops are more representatives for countries with a warm climate such as Greece, Spain, Cyprus, and Portugal. The largest share of arable land is owned by Finland (Figure 2).

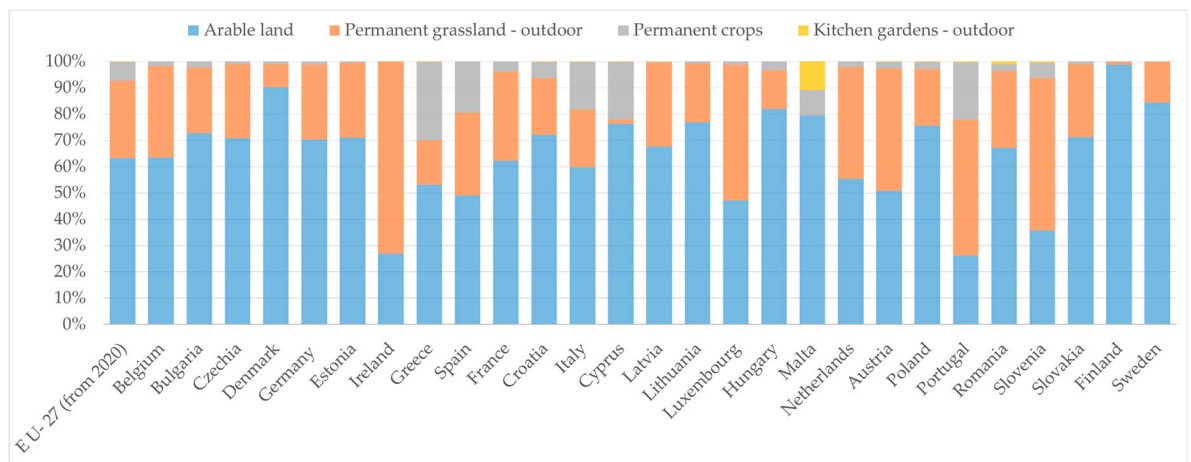

**Figure 2.** European Union UAA structure. Source: Authors' calculation and representation, according to Eurostat (2020) data.

The extensive areas intended for agriculture as well as the variety of available land categories can ensure the development of a certain type of agriculture specific to the existing foundation in order to optimally exploit the resources. Therefore, the comparative analysis between countries must also take into account the aspects of distribution on the globe, location relief, and climatic conditions.

One of the indicators for which a significant value does not necessarily mean a high level of performance is represented by the number of farms/agricultural holdings that each country owns. Within the EU, the largest percentage is owned by Romania (31.8%), Poland (14.4%), and Italy (12.5%). The largest area of UAA for each farm, on average, is owned by Czechia with 120.9 ha followed by Slovakia (95.1 ha) and the largest economic size per farm is recorded in the Netherlands (EUR 448,100), at the level of EU 27 the average being EUR 38700. Romania registers the lowest value of this indicator (EUR 4100) due to the very large number of farms generated by the excessive fragmentation of agricultural land (Table 2).

**Table 2.** Main indicators regarding farms and their economic size for EU, 2020.

| No. | Country/Item | Farm Number | | Total Economic Size | | UAA/Farm (ha) | Economic Size/Farm (Thousand Euro/S.O.) |
|---|---|---|---|---|---|---|---|
| | | Number (Thousands) | % of EU | Value (Millions Euro/S.O.) | % of EU | | |
| 1 | Austria | 110.8 | 1.2 | 6330.0 | 1.8 | 23.5 | 57.1 |
| 2 | Belgium | 36.0 | 0.4 | 8083.5 | 2.3 | 38.0 | 224.5 |
| 3 | Bulgaria | 132.7 | 1.5 | 3772.0 | 1.1 | 34.4 | 28.4 |
| 4 | Croatia | 143.9 | 1.6 | 1945.5 | 0.6 | 10.5 | 13.5 |
| 5 | Cyprus | 34.1 | 0.4 | 749.3 | 0.2 | 3.9 | 22.0 |
| 6 | Czechia | 28.9 | 0.3 | 5224.7 | 1.5 | 120.9 | 180.8 |
| 7 | Denmark | 37.1 | 0.4 | 8886.7 | 2.5 | 70.9 | 239.5 |
| 8 | Estonia | 11.4 | 0.1 | 756.5 | 0.2 | 85.5 | 66.4 |
| 9 | Finland | 45.6 | 0.5 | 3192.4 | 0.9 | 50.0 | 70.0 |
| 10 | France | 393.0 | 4.3 | 62,267.6 | 17.7 | 69.6 | 158.4 |
| 11 | Germany | 262.8 | 2.9 | 44,050.0 | 12.5 | 63.1 | 167.6 |
| 12 | Greece | 530.8 | 5.9 | 7354.9 | 2.1 | 7.4 | 13.9 |
| 13 | Hungary | 232.1 | 2.6 | 6387.9 | 1.8 | 21.2 | 27.5 |
| 14 | Ireland | 130.2 | 1.4 | 6851.2 | 2.0 | 37.8 | 52.6 |
| 15 | Italy | 1133.0 | 12.5 | 65,353.4 | 18.6 | 11.1 | 57.7 |
| 16 | Latvia | 69.0 | 0.8 | 1319.6 | 0.4 | 28.5 | 19.1 |

| No. | Country/Item | Farm Number | | Total Economic Size | | UAA/Farm (ha) | Economic Size/Farm (Thousand Euro/S.O.) |
|-----|--------------|-------------|--------|---------------------|--------|-------------|-------------------|
| | | Number (Thousands) | % of EU | Value (Millions Euro/S.O.) | % of EU | | |
| 17 | Lithuania | 132.1 | 1.5 | 2215.6 | 0.6 | 22.1 | 16.8 |
| 18 | Luxembourg | 1.9 | 0.0 | 323.5 | 0.1 | 69.5 | 170.3 |
| 19 | Malta | 7.7 | 0.1 | 66.9 | 0.0 | 1.3 | 8.7 |
| 20 | Netherlands | 52.6 | 0.6 | 23,571.2 | 6.7 | 34.6 | 448.1 |
| 21 | Poland | 1302.3 | 14.4 | 25,630.6 | 7.3 | 11.4 | 19.7 |
| 22 | Portugal | 290.2 | 3.2 | 6369.1 | 1.8 | 13.7 | 21.9 |
| 23 | Romania | 2887.1 | 31.8 | 11,692.8 | 3.3 | 4.4 | 4.1 |
| 24 | Slovakia | 19.6 | 0.2 | 1836.2 | 0.5 | 95.1 | 93.7 |
| 25 | Slovenia | 72.5 | 0.8 | 1120.9 | 0.3 | 6.7 | 15.5 |
| 26 | Spain | 914.9 | 10.1 | 40,368.2 | 11.5 | 26.1 | 44.1 |
| 27 | Sweden | 58.8 | 0.6 | 5359.1 | 1.5 | 51.1 | 91.1 |
| 28 | EU 27 (from 2020) | 9071.0 | 100.0 | 351,079.3 | 100.0 | 17.4 | 38.7 |

Source: Authors' calculation, according to Eurostat (2020) data.

One of the most important determinants of the economic performance in general and of agriculture in particular is gross fixed capital formation (GFCF). This indicator reveals and anticipates significant increases in terms of agricultural output because increasing the performance of machines and equipment used in the production process implicitly leads to an increase in the sector's performance as a whole in the near future knowing that innovation generates opportunities, added value, and development in general, even a small technological change can have substantial effect on the economic growth [65]. GFCF in agriculture, on average for EU 27 represents 2.3% of total GDP while GFCF for all sectors represents 22.1%. GFCF in machines and other agricultural equipment is 44.3% of agricultural GFCF with a value of EUR 24902.79 million, the country with the largest share being Latvia (74.4 %) (Table 3).

**Table 3.** Gross fixed capital formation-EU, 2020.

| No. | Country/Item | Total Economy | | Agriculture | | Machines and Other Agricultural Equipment | |
|-----|--------------|---------------|----------|-------------|------------------|-----------------------------------------|------------------------|
| | | Value (Million Euro) | % of GDP | Value (Million Euro) | % of Total Economy | Value (Million Euro) | % of Agricultural GFCF |
| 1 | Austria | 95,768.40 | 25.2 | 2251.38 | 2.52 | 995.00 | 44.2 |
| 2 | Belgium | 109,301.80 | 23.9 | 1327.63 | 1.43 | 315.39 | 23.8 |
| 3 | Bulgaria | 11,750.20 | 19.2 | 436.33 | 4.84 | 211.71 | 48.5 |
| 4 | Croatia | 11,197.50 | 22.3 | 201.40 | 3.92 | 37.28 | 18.5 |
| 5 | Cyprus | 4308.30 | 19.9 | 34.47 | 0.68 | 1.48 | 4.3 |
| 6 | Czechia | 57,290.80 | 26.5 | 972.94 | 3.00 | 402.64 | 41.4 |
| 7 | Denmark | 69,506.20 | 22.3 | 1217.46 | 1.55 | 711.28 | 58.4 |
| 8 | Estonia | 8233.10 | 30.7 | 213.01 | 2.64 | 128.38 | 60.3 |
| 9 | Finland | 57,463.00 | 24.1 | 1186.00 | 2.87 | 204.00 | 17.2 |
| 10 | France | 529,854.00 | 22.9 | 10,317.77 | 2.19 | 6556.33 | 63.5 |
| 11 | Germany | 735,869.00 | 21.9 | 9342.96 | 1.29 | 5698.00 | 61.0 |
| 12 | Greece | 19,271.00 | 11.7 | 1792.99 | 9.84 | 904.46 | 50.4 |
| 13 | Hungary | 36,602.90 | 26.6 | 1148.20 | 3.54 | 287.45 | 25.0 |
| 14 | Ireland | 158,065.80 | 42.4 | 1127.24 | 0.79 | 606.19 | 53.8 |
| 15 | Italy | 295,658.80 | 17.8 | 8262.10 | 3.11 | 2377.51 | 28.8 |
| 16 | Latvia | 7217.00 | 24.5 | 243.12 | 6.03 | 180.97 | 74.4 |
| 17 | Lithuania | 10,423.50 | 21.1 | 543.30 | 5.92 | 208.00 | 38.3 |
| 18 | Luxembourg | 10,788.60 | 16.8 | 99.55 | 1.41 | 30.81 | 30.9 |
| 19 | Malta | 2686.10 | 20.6 | 11.78 | 0.67 | 2.76 | 23.4 |
| 20 | Netherlands | 172,937.00 | 21.7 | 4984.88 | 2.99 | 2390.73 | 48.0 |
| 21 | Poland | 90,667.80 | 17.2 | 1370.46 | 3.80 | 485.44 | 35.4 |
| 22 | Portugal | 38,183.10 | 19.1 | 1129.79 | 3.47 | 277.73 | 24.6 |
| 23 | Romania | 52,182.60 | 23.8 | 1612.95 | 4.19 | 333.21 | 20.7 |
| 24 | Slovakia | 18,072.90 | 19.6 | 132.89 | 3.69 | 27.76 | 20.9 |
| 25 | Slovenia | 8860.50 | 18.9 | 236.44 | 4.51 | 70.32 | 29.7 |
| 26 | Spain | 227,599.00 | 20.3 | 4823.95 | 2.92 | 908.92 | 18.8 |
| 27 | Sweden | 120,694.30 | 25.1 | 1148.45 | 2.14 | 549.05 | 47.8 |
| 28 | EU 27 (from 2020) | 2,960,453.50 | 22.1 | 56,169.44 | 2.30 | 24,902.79 | 44.3 |

Source: Authors' calculation, according to Eurostat (2020) data.

Gross fixed capital formation in agriculture plays a fundamental role in the development strategy of each farm and implicitly of each country. The level of total investments in agricultural equipment, buildings or other fixed capital, GFCF per farm or GFCF per each of agricultural land are indicators of the performance of this sector, which reflect much more clearly the level of development but also the orientation of the activities in the field.

The EU 27 recorded in 2020, on average EUR 2073.7 million in gross fixed capital formation in agriculture, the country with the highest value being France. GFCF per farm, on average for the EU 27 (from 2020) members was EUR 6192.2 per hectare with the highest value recorded by the Netherlands (94697.6 euros/farm) and the lowest value by Romania (558.7 euros/farm)—a fact determined by the very large number of existing farms in relation to the other member countries. GFCF per hectare of UAA for the EU 27 was 358 euro/ha. The leader of agricultural performance in terms of this indicator is also the Netherlands with 2742.1 euro/ha and the country with the lowest value of this derived indicator is Slovakia (71.3 euro/ha) (Table 4)

**Table 4.** Gross fixed capital formation per farm and per hectare of UAA-EU, 2020.

| No. | Country/Item | GFCF—Total Value (Million Euro) | GFCF/Farm (Euro/Farm) | GFCF/UAA (Euro/ha) |
|---|---|---|---|---|
| 1 | Austria | 2251.38 | 20,323.0 | 865.0 |
| 2 | Belgium | 1327.63 | 36,878.6 | 970.4 |
| 3 | Bulgaria | 436.33 | 3287.1 | 95.6 |
| 4 | Croatia | 201.40 | 1399.3 | 133.8 |
| 5 | Cyprus | 34.47 | 1012.3 | 257.0 |
| 6 | Czechia | 972.94 | 33,654.1 | 278.6 |
| 7 | Denmark | 1217.46 | 32,824.5 | 462.9 |
| 8 | Estonia | 213.01 | 18,734.4 | 218.4 |
| 9 | Finland | 1186.00 | 25,991.7 | 519.8 |
| 10 | France | 10,317.77 | 26,251.9 | 377.0 |
| 11 | Germany | 9342.96 | 35,554.3 | 563.0 |
| 12 | Greece | 1792.99 | 3378.2 | 457.8 |
| 13 | Hungary | 1148.20 | 4947.9 | 233.3 |
| 14 | Ireland | 1127.24 | 8656.4 | 229.1 |
| 15 | Italy | 8262.10 | 7292.1 | 659.7 |
| 16 | Latvia | 243.12 | 3524.5 | 123.5 |
| 17 | Lithuania | 543.30 | 4113.4 | 186.4 |
| 18 | Luxembourg | 99.55 | 52,952.1 | 753.4 |
| 19 | Malta | 11.78 | 1539.9 | 1202.0 |
| 20 | Netherlands | 4984.88 | 94,697.6 | 2742.1 |
| 21 | Poland | 1370.46 | 1052.3 | 92.7 |
| 22 | Portugal | 1129.79 | 3892.7 | 285.0 |
| 23 | Romania | 1612.95 | 558.7 | 126.4 |
| 24 | Slovakia | 132.89 | 6769.7 | 71.3 |
| 25 | Slovenia | 236.44 | 3262.6 | 489.1 |
| 26 | Spain | 4823.95 | 5272.8 | 201.7 |
| 27 | Sweden | 1148.45 | 19,534.8 | 382.1 |
| 28 | EU 27 (from 2020) | 56,169.44 | 6192.2 | 356.8 |

Source: Authors' calculation, according to Eurostat (2020) data.

Farmers receive benefits derived from the Common Agricultural Policy [66], benefits that also contribute to the general performance of agricultural activity and the general performance of the sector. Differences between countries in terms of support given to farmers can influence the overall results and the strategies adopted by each individual Member State.

For most EU farmers, subsidies and especially direct payments play an important role in ensuring the viability and performance of agricultural activity. In 2020, a total of EUR 38.2 billion in direct payments was recorded, with an average of EUR 1.4 billion for each Member State (Table 5).

**Table 5.** CAP expenditure by Member State in 2020, thousand euro.

| No. | Country/Item | Direct Payments | Market Measures | Rural Development | Total |
|---|---|---|---|---|---|
| 1 | Austria | 691,597 | 22,298 | 567,266 | 1281,161 |
| 2 | Belgium | 481,836 | 60,758 | 102,723 | 645,317 |
| 3 | Bulgaria | 781,855 | 18,386 | 338,990 | 1,139,231 |
| 4 | Croatia | 317,338 | 13,061 | 282,343 | 612,741 |
| 5 | Cyprus | 48,125 | 5922 | 18,881 | 72,929 |
| 6 | Czechia | 855,832 | 16,537 | 321,615 | 1,193,984 |
| 7 | Denmark | 814,070 | 12,212 | 151,589 | 977,871 |
| 8 | Estonia | 142,536 | 1476 | 129,177 | 273,189 |
| 9 | Finland | 523,450 | 6473 | 344,777 | 874,699 |
| 10 | France | 6,909,823 | 550,551 | 1,987,740 | 9,448,114 |
| 11 | Germany | 4,768,123 | 117,256 | 1,394,589 | 6,279,967 |
| 12 | Greece | 1,982,609 | 59,445 | 698,261 | 2,740,315 |
| 13 | Hungary | 1,266,719 | 40,211 | 486,663 | 1,793,593 |
| 14 | Ireland | 1,201,194 | 59,338 | 312,570 | 1,573,102 |
| 15 | Italy | 3,599,133 | 677,514 | 1,501,763 | 5,778,411 |
| 16 | Latvia | 277,306 | 3048 | 161,492 | 441,846 |
| 17 | Lithuania | 480,492 | 3344 | 264,151 | 747,987 |
| 18 | Luxembourg | 32,841 | 556 | 14,511 | 47,909 |
| 19 | Malta | 5117 | 344 | 13,859 | 19,320 |
| 20 | Netherlands | 666,190 | 22,583 | 147,976 | 836,749 |
| 21 | Poland | 3,402,201 | 25,553 | 1,187,301 | 4,615,055 |
| 22 | Portugal | 680,228 | 107,898 | 582,456 | 1,370,581 |
| 23 | Romania | 1,912,461 | 65,671 | 1,139,927 | 3,118,059 |
| 24 | Slovakia | 447,758 | 11,255 | 214,525 | 673,538 |
| 25 | Slovenia | 133,869 | 7022 | 120,721 | 261,611 |
| 26 | Spain | 5,125,093 | 599,856 | 1,183,394 | 6,908,343 |
| 27 | Sweden | 686,818 | 11,875 | 249,819 | 948,511 |
| 28 | EU 27 (from 2020) | 38,234,612 | 2,520,441 | 13,919,080 | 54,674,132 |

Source: [66].

Rural development is also a pillar of the Common Agricultural Policy which has gained more and more importance at the level of the member countries [67]. A total of EUR 13919 million was allocated for these rural development measures, including extensive activities and not just the objectives regarding agricultural production.

Regarding the share of direct payments in agricultural output at the level of the year 2020 for EU 27, an average percentage of 10.7% is calculated, with Bulgaria having the highest value (20.6%). The most representative value in highlighting the impact of direct payments on the overall result is the percentage of direct payments in factor income. For the European Union, on average, this percentage is around 30%. According to the statistics analyzed and the processing of the values identified in Sweden for farmers, the value of these direct payments is more than half of the income factor obtained (52.11%), while Netherland's farmers stand for 9.38% (Table 6).

**Table 6.** Direct payments (euro/ha) and share of direct payments in income (%).

| No. | Country/Item | Direct Payments Euro/ha | | % of Direct Payments in Agricultural Output/UAA (Including Subsidies) | % of Direct Payments in Factor Income |
|---|---|---|---|---|---|
| | | Value | Deviation | | |
| 1 | Austria | 350 | 34.4 | 10.6 | 40.51 |
| 2 | Belgium | 377 | 61.4 | 5.4 | 20.64 |
| 3 | Bulgaria | 229 | −86.6 | 20.6 | 44.62 |
| 4 | Croatia | 365 | 49.4 | 18.5 | 40.72 |
| 5 | Cyprus | 377 | 61.4 | 6.2 | 12.40 |
| 6 | Czechia | 273 | −42.6 | 14.5 | 41.12 |
| 7 | Denmark | 308 | −7.6 | 6.5 | 26.34 |
| 8 | Estonia | 170 | −145.6 | 14.3 | 48.37 |
| 9 | Finland | 302 | −13.6 | 13.3 | 39.21 |
| 10 | France | 291 | −24.6 | 9.4 | 30.63 |

**Table 6.** *Cont.*

| No. | Country/Item | Direct Payments Euro/ha | | % of Direct Payments in Agricultural Output/UAA (Including Subsidies) | % of Direct Payments in Factor Income |
|---|---|---|---|---|---|
| | | Value | Deviation | | |
| 11 | Germany | 283 | −32.6 | 7.6 | 30.12 |
| 12 | Greece | 546 | 230.4 | 15.1 | 29.64 |
| 13 | Hungary | 254 | −61.6 | 13.0 | 33.40 |
| 14 | Ireland | 239 | −76.6 | 11.7 | 29.33 |
| 15 | Italy | 379 | 63.4 | 7.6 | 18.64 |
| 16 | Latvia | 166 | −149.6 | 15.9 | 43.86 |
| 17 | Lithuania | 176 | −139.6 | 12.8 | 38.51 |
| 18 | Luxembourg | 318 | 2.4 | 8.7 | 44.36 |
| 19 | Malta | 911 | 595.4 | 6.9 | 12.24 |
| 20 | Netherlands | 369 | 53.4 | 2.3 | 9.38 |
| 21 | Poland | 246 | −69.6 | 12.1 | 30.26 |
| 22 | Portugal | 264 | −51.6 | 11.1 | 32.44 |
| 23 | Romania | 215 | −100.6 | 14.0 | 44.10 |
| 24 | Slovakia | 231 | −84.6 | 15.5 | 48.17 |
| 25 | Slovenia | 371 | 55.4 | 11.6 | 31.17 |
| 26 | Spain | 254 | −61.6 | 10.5 | 22.31 |
| 27 | Sweden | 258 | −57.6 | 11.2 | 52.11 |
| 28 | EU 27 (from 2020) average | 315.6 | - | 10.7 | 29.90 |

Source: Authors' calculation, according to Eurostat (2020) data.

For the data represented, the mean value of the Direct payments per ha is EUR 315.6 per hectare of utilized agricultural area, with a standard deviation for the values by 144.7, a variance of 20944.1, the coefficient of variation being 45.9.

Direct payments (euro/ha) have also a direct influence on factor income/UAA that farmers obtain as a result of their activity (Figure 3).

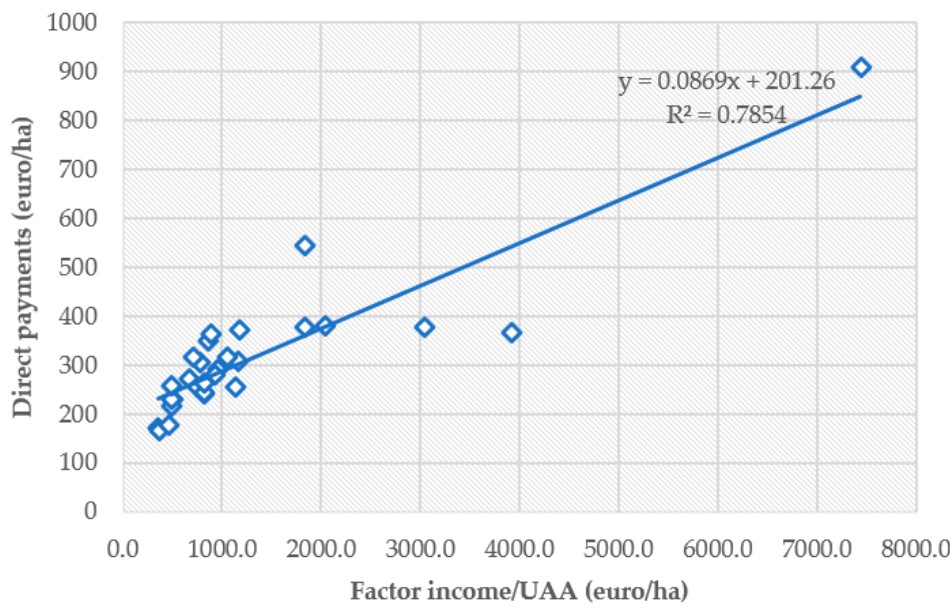

**Figure 3.** Correlation between factor income/UAA and direct payments (euro/ha).

Having $R^2 = 0.7854$ determines a Pearson Correlation Coefficient $R = \sqrt{0.7854} = 0.88622$ which means a pozitive correlation between the two variables, the 27 EU countries (represented by blue squares within the figure above) having a balanced distribution in relation to the trendline.

In order to analyze one of the established hypotheses, the variables gross fixed capital formation (million euros) and total agricultural output (million euros) were taken into account to highlight the connection between them (Figure 4).

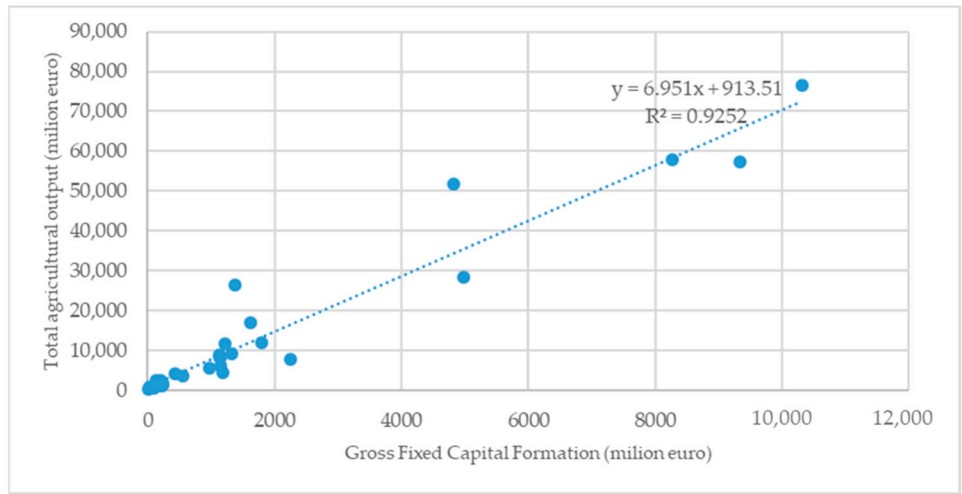

**Figure 4.** Correlation between gross fixed capital formation and total agricultural output.

The R$^2$ value on chart is 0.9252 allowing calculation of Pearson Correlation Coefficient R = $\sqrt{0.9252}$ = 0.96187 which means a very strong positive correlation between the two variables. This result validates the first established hypothesis that gross fixed capital formation has a direct positive influence on total agricultural output, being an indirect indicator of agricultural performance.

It is known that an effective combination of capital with other production factors, such as labor and land, determine higher outputs and as a consequence higher income. [55].

In the performance analysis, in addition to the results regarding the value of production, gross fixed capital formation or subsidies, labor productivity are also included. The most efficient use of labor in the conditions of an increasingly sensitive situation of the work force in agriculture but also in other sectors of the economy, is one of the strategic directions targeted at the European level through the related agricultural policies.

The value of annual working units for the EU in 2020 was 7959.72 with an 0.9 AWU/farm on average, the highest labor productivity being recorded by Czechia (3.3 AWU/farm). Total agricultural output for the EU in 2020 was EUR 415098.18 million, France having the largest share (18.4%) and Malta the lowest, at EUR 121.09 million (0.02%). In order to analyze the utilization degree of labor more deeply, the derived indicator total agricultural output per AWU was calculated, thus identifying Denmark as the country with the highest level of the indicator (227,164.6 Euro/AWU), the average at the level of the Member States being (52,149.8 Euro AWU) (Table 7).

**Table 7.** EU 2020 Agricultural labor input statistics: AWU (thousand), agricultural output (million euro).

| No. | Country/Item | AWU | | Total Agricultural Output | | |
| --- | --- | --- | --- | --- | --- | --- |
| | | Total Value | AWU/ Farm | Total Value | Euro/ Farm | Euro/ AWU |
| 1 | Austria | 121.57 | 1.1 | 7645.59 | 69,003.5 | 62,890.4 |
| 2 | Belgium | 52.21 | 1.5 | 9057.99 | 251,610.8 | 173,491.5 |
| 3 | Bulgaria | 181.90 | 1.4 | 4021.96 | 30,308.7 | 22,110.8 |
| 4 | Croatia | 172.28 | 1.2 | 2422.11 | 16,831.9 | 14,059.1 |
| 5 | Cyprus | 19.13 | 0.6 | 759.14 | 22,262.2 | 39,683.2 |
| 6 | Czechia | 95.37 | 3.3 | 5632.71 | 194,903.5 | 59,061.7 |
| 7 | Denmark | 51.14 | 1.4 | 11,617.20 | 313,132.1 | 227,164.6 |
| 8 | Estonia | 17.28 | 1.5 | 995.34 | 87,310.5 | 57,600.7 |
| 9 | Finland | 65.59 | 1.4 | 4475.27 | 98,141.9 | 68,231.0 |
| 10 | France | 710.21 | 1.8 | 76,630.72 | 194,989.1 | 107,898.7 |

**Table 7.** *Cont.*

| No. | Country/Item | AWU | | Total Agricultural Output | | |
|---|---|---|---|---|---|---|
| | | Total Value | AWU/ Farm | Total Value | Euro/ Farm | Euro/ AWU |
| 11 | Germany | 469.00 | 1.8 | 57,345.40 | 218,209.3 | 122,271.6 |
| 12 | Greece | 326.40 | 0.6 | 12,051.74 | 22,704.9 | 36,923.2 |
| 13 | Hungary | 326.94 | 1.4 | 8398.47 | 36,184.7 | 25,688.1 |
| 14 | Ireland | 156.94 | 1.2 | 8891.82 | 68,293.5 | 56,657.4 |
| 15 | Italy | 1059.30 | 0.9 | 57,833.34 | 51,044.4 | 54,595.8 |
| 16 | Latvia | 67.15 | 1.0 | 1727.16 | 25,031.3 | 25,720.9 |
| 17 | Lithuania | 125.26 | 0.9 | 3486.40 | 26,392.1 | 27,833.3 |
| 18 | Luxembourg | 3.55 | 1.9 | 439.85 | 231,500.0 | 123,901.4 |
| 19 | Malta | 5.40 | 0.7 | 121.09 | 15,726.0 | 22,424.1 |
| 20 | Netherlands | 156.70 | 3.0 | 28,235.59 | 536,798.3 | 180,188.8 |
| 21 | Poland | 1427.70 | 1.1 | 26,405.78 | 20,276.3 | 18,495.3 |
| 22 | Portugal | 233.36 | 0.8 | 8403.49 | 28,957.6 | 36,010.8 |
| 23 | Romania | 1090.00 | 0.4 | 16,824.17 | 5827.4 | 15,435.0 |
| 24 | Slovakia | 42.50 | 2.2 | 2348.02 | 119,796.9 | 55,247.5 |
| 25 | Slovenia | 74.05 | 1.0 | 1370.29 | 18,900.6 | 18,504.9 |
| 26 | Spain | 851.38 | 0.9 | 51,787.23 | 56,604.3 | 60,827.4 |
| 27 | Sweden | 57.41 | 1.0 | 6170.33 | 104,937.6 | 107,478.3 |
| 28 | EU 27 (from 2020) | 7959.72 | 0.9 | 415,098.18 | 45,761.0 | 52,149.8 |

Source: Authors' calculation, according to Eurostat (2020) data.

In order to have a more accurate image of the efficiency of the use of labor, the AWU was divided by UAA, the average for all the countries being 0.05 per hectare with the highest value for Malta (0.55), as the country with a small value of UAA. Moreover, for one hectare of UAA there are associated EUR 2637.0 of Agricultural output on average for all the countries; Netherlands dominating this top with 15532.0 Euro/UAA, Latvia and Bulgaria being the countries with the lowest score for the indicator taken into account (under 1000 euro/UAA).

A new way of comparing the efficiency of agricultural activity is generated by the derived indicator calculated as GFGF per UAA, when placed next to the previous indicator helps to identify the concrete situation based on the actual cultivated agricultural land. The value of this indicator was 865.0 euro/UAA for Austria, 659.7 euro/UAA for Italy, and 71.3 euro/UAA for Slovakia, with an EU 27 average of 356.8 euro/UAA regarding gross fixed capital formation (Table 8).

**Table 8.** The connection between agricultural labor, total agricultural output, and GFGF-EU, 2020.

| No. | Country/Item | AWU | | Total Agricultural Output | | Gross Fixed Capital Formation | |
|---|---|---|---|---|---|---|---|
| | | Total Value (Thousand) | AWU/ UAA | Total Value (Million Euro) | Euro/ UAA | Total Value (Million Euro) | Euro/ UAA |
| 1 | Austria | 121.57 | 0.05 | 7645.59 | 2937.6 | 2251.38 | 865.0 |
| 2 | Belgium | 52.21 | 0.04 | 9057.99 | 6620.8 | 1327.63 | 970.4 |
| 3 | Bulgaria | 181.90 | 0.04 | 4021.96 | 881.2 | 436.33 | 95.6 |
| 4 | Croatia | 172.28 | 0.11 | 2422.11 | 1608.9 | 201.40 | 133.8 |
| 5 | Cyprus | 19.13 | 0.14 | 759.14 | 5659.3 | 34.47 | 257.0 |
| 6 | Czechia | 95.37 | 0.03 | 5632.71 | 1612.8 | 972.94 | 278.6 |
| 7 | Denmark | 51.14 | 0.02 | 11,617.20 | 4417.3 | 1217.46 | 462.9 |
| 8 | Estonia | 17.28 | 0.02 | 995.34 | 1020.5 | 213.01 | 218.4 |
| 9 | Finland | 65.59 | 0.03 | 4475.27 | 1961.4 | 1186.00 | 519.8 |
| 10 | France | 710.21 | 0.03 | 76,630.72 | 2800.4 | 10,317.77 | 377.0 |
| 11 | Germany | 469.00 | 0.03 | 57,345.40 | 3455.6 | 9342.96 | 563.0 |
| 12 | Greece | 326.40 | 0.08 | 12,051.74 | 3077.1 | 1792.99 | 457.8 |

**Table 8.** *Cont.*

| No. | Country/Item | AWU | | Total Agricultural Output | | Gross Fixed Capital Formation | |
|-----|--------------|-----|-----|-----|-----|-----|-----|
| | | Total Value (Thousand) | AWU/ UAA | Total Value (Million Euro) | Euro/ UAA | Total Value (Million Euro) | Euro/ UAA |
| 13 | Hungary | 326.94 | 0.07 | 8398.47 | 1706.4 | 1148.20 | 233.3 |
| 14 | Ireland | 156.94 | 0.03 | 8891.82 | 1807.2 | 1127.24 | 229.1 |
| 15 | Italy | 1059.30 | 0.08 | 57,833.34 | 4618.0 | 8262.10 | 659.7 |
| 16 | Latvia | 67.15 | 0.03 | 1727.16 | 877.2 | 243.12 | 123.5 |
| 17 | Lithuania | 125.26 | 0.04 | 3486.40 | 1196.2 | 543.30 | 186.4 |
| 18 | Luxembourg | 3.55 | 0.03 | 439.85 | 3328.7 | 99.55 | 753.4 |
| 19 | Malta | 5.40 | 0.55 | 121.09 | 12,356.1 | 11.78 | 1202.0 |
| 20 | Netherlands | 156.70 | 0.09 | 28,235.59 | 15,532.0 | 4984.88 | 2742.1 |
| 21 | Poland | 1427.70 | 0.10 | 26,405.78 | 1786.1 | 1370.46 | 92.7 |
| 22 | Portugal | 233.36 | 0.06 | 8403.49 | 2120.0 | 1129.79 | 285.0 |
| 23 | Romania | 1090.00 | 0.09 | 16,824.17 | 1318.2 | 1612.95 | 126.4 |
| 24 | Slovakia | 42.50 | 0.02 | 2348.02 | 1260.6 | 132.89 | 71.3 |
| 25 | Slovenia | 74.05 | 0.15 | 1370.29 | 2834.5 | 236.44 | 489.1 |
| 26 | Spain | 851.38 | 0.04 | 51,787.23 | 2165.6 | 4823.95 | 201.7 |
| 27 | Sweden | 57.41 | 0.02 | 6170.33 | 2052.8 | 1148.45 | 382.1 |
| 28 | EU 27 (from 2020) | 7959.72 | 0.05 | 415,098.18 | 2637.0 | 56,169.44 | 356.8 |

Source: Authors' calculation, according to Eurostat (2020) data.

One of the most significant challenges for agriculture is represented by sustainable efficiency enhancement of agricultural production [68]. In order to have an overview of the value of agricultural production, we can use its structural analysis. Total output is formed not only by crop output and animal output, but also by agricultural services output and secondary activities. For the EU 27, in 2020, the largest amount in total agricultural output is owned by crop output (EUR 221,015.36 million), France contributing with EUR 42,670.67 million (Table 9), the structure differing from one country to another depending on the utilization of available resources, tradition, and own development strategies.

**Table 9.** Total agricultural output value structure (million euro).

| No. | Country/Item | Crop Output | Animal Output | Agricultural Services Output | Secondary Activities | Total Output |
|-----|--------------|-------------|---------------|------------------------------|----------------------|--------------|
| 1 | Austria | 3324.96 | 3582.74 | 290.38 | 447.51 | 7645.59 |
| 2 | Belgium | 4110.79 | 4711.29 | 199.43 | 36.48 | 9057.99 |
| 3 | Bulgaria | 2678.21 | 1003.17 | 236.24 | 104.34 | 4021.96 |
| 4 | Croatia | 1434.10 | 832.76 | 94.52 | 60.73 | 2422.11 |
| 5 | Cyprus | 285.56 | 455.65 | 0.37 | 17.56 | 759.14 |
| 6 | Czechia | 3302.55 | 1988.06 | 151.17 | 190.93 | 5632.71 |
| 7 | Denmark | 3888.43 | 6957.68 | 599.96 | 171.13 | 11,617.20 |
| 8 | Estonia | 484.27 | 414.87 | 63.47 | 32.73 | 995.34 |
| 9 | Finland | 1556.81 | 2286.14 | 135.54 | 496.78 | 4475.27 |
| 10 | France | 42,670.67 | 26,458.82 | 5030.30 | 2470.93 | 76,630.72 |
| 11 | Germany | 27,528.47 | 26,416.83 | 2432.78 | 967.32 | 57,345.40 |
| 12 | Greece | 8504.76 | 2353.60 | 304.98 | 888.40 | 12,051.74 |
| 13 | Hungary | 4939.01 | 2858.23 | 469.78 | 131.45 | 8398.47 |
| 14 | Ireland | 1927.02 | 6534.02 | 430.78 | 0.00 | 8891.82 |
| 15 | Italy | 32,824.79 | 15,506.83 | 4914.08 | 4587.64 | 57,833.34 |
| 16 | Latvia | 1029.04 | 533.19 | 34.74 | 130.19 | 1727.16 |
| 17 | Lithuania | 2167.28 | 930.18 | 42.60 | 346.34 | 3486.40 |
| 18 | Luxembourg | 152.19 | 248.98 | 3.21 | 35.47 | 439.85 |
| 19 | Malta | 43.55 | 70.72 | 0.00 | 6.82 | 121.09 |
| 20 | Netherlands | 13,952.68 | 10,634.50 | 2817.38 | 831.03 | 28,235.59 |
| 21 | Poland | 12,925.06 | 12,799.27 | 592.36 | 89.09 | 26,405.78 |
| 22 | Portugal | 4914.59 | 2992.37 | 235.94 | 260.59 | 8403.49 |
| 23 | Romania | 10,914.36 | 4047.46 | 379.56 | 1482.79 | 16,824.17 |
| 24 | Slovakia | 1288.88 | 755.29 | 165.47 | 138.38 | 2348.02 |
| 25 | Slovenia | 790.65 | 546.39 | 33.25 | 0.00 | 1370.29 |
| 26 | Spain | 30,484.60 | 19,732.35 | 609.93 | 960.35 | 51,787.23 |
| 27 | Sweden | 2892.09 | 2754.47 | 378.43 | 145.34 | 6170.33 |
| 28 | EU 27 (from 2020) | 221,015.36 | 158,405.87 | 20,646.67 | 15,030.28 | 415,098.18 |

Source: Authors' calculation, according to Eurostat (2020) data.

The output structure for each country is not only consistent with the related UAA structure but also with the development level of the sectors adjacent to the actual agricultural production and of the processing sector (which immediately follows the production sector). Regarding the share of agricultural services output in the total output for the EU 27, in 2020 it amounted to 4.9%, the leading countries being Netherlands with 9.9%, Italy, (8.5%) and Slovakia (7.5%).

At the level of the EU, crop output represents 53% of total agricultural output, followed by animal output with 38% (Figure 5).

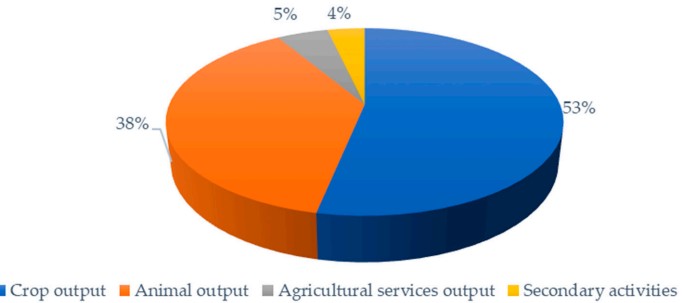

**Figure 5.** Total agricultural output structure at the level of the EU, 2020. Source: Authors' calculation and representation, according to Eurostat (2020) data.

The largest shares of crop output value, in total output value, are held by Greece (70.5%), Bulgaria (66.5%), and Romania (64.8%), while for animal output the leaders are Ireland (73.4%), Cyprus (60%) and Denmark (59.8%), the countries that mainly develop a certain sector (crop or animal) also being the ones that are last for the related complementary sector (Figure 6).

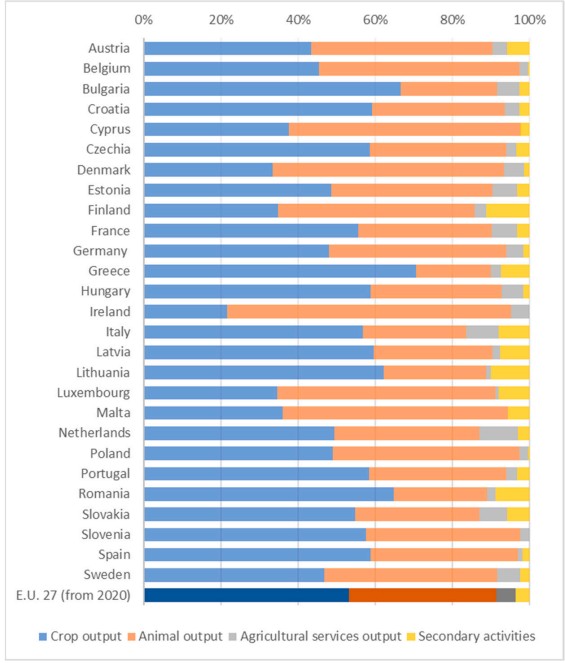

**Figure 6.** The total agricultural output structure for each member state, 2020. Source: Authors' calculation and representation, according to Eurostat (2020) data.

In addition to the other sectors, modern agriculture and the processing sectors immediately following the production process have determined an increase in the GVA volume in the economy [69]. In order to assess the agricultural performance, it is not enough to identify the value of the output obtained. Another essential indicator that reflects much

more accurately the performance of agricultural activity within a farm is factor income which is calculated as follows: gross value added − fixed capital consumption − other taxes on production + other subsidies on production.

Factor income (agricultural income) for EU 27 was EUR 166.2 billion in 2020, resulting from a gross value added (total agricultural output − intermediate consumption) of EUR 178.4 billion (Table 10). The country that recorded the highest level of total factor income is Spain with EUR 27,231.24 million, followed by France (EUR 25,994.09 million) and Italy (EUR 25,466.58 million). Regarding the derived indicator (factor income per UAA) which more clearly reflects the reality of agricultural efficiency, Malta, the Netherlands, and Cyprus lead this ranking while Estonia, Latvia, and Lithuania rank last.

**Table 10.** Factor income determination using specific indicators (million euro).

| No. | Country/Item | Gross Value Added (GVA) | Total Intermediate Consumption | Fixed Capital Consumption | Other Taxes on Production | Other Subsidies on Production | Factor Income | Factor Income/UAA |
|---|---|---|---|---|---|---|---|---|
| 1 | Austria | 2472.3 | 6504.04 | 812.30 | 2.84 | 622.62 | 2249.16 | 864.1 |
| 2 | Belgium | 3122.3 | 4410.47 | 1878.16 | 155.95 | 1488.53 | 2498.79 | 1826.6 |
| 3 | Bulgaria | 1748.4 | 2181.84 | 453.27 | 7.70 | 1125.64 | 2342.56 | 513.3 |
| 4 | Croatia | 1153.2 | 1291.14 | 316.27 | 0.00 | 491.92 | 1349.12 | 896.4 |
| 5 | Cyprus | 362.3 | 401.38 | 15.79 | 10.07 | 66.85 | 407.37 | 3040.1 |
| 6 | Czechia | 1935.1 | 3652.94 | 773.94 | 43.20 | 1224.29 | 2318.90 | 663.9 |
| 7 | Denmark | 3567.8 | 7832.34 | 1141.14 | 138.39 | 883.15 | 3075.26 | 1169.3 |
| 8 | Estonia | 256.7 | 742.55 | 150.57 | 3.93 | 239.08 | 342.68 | 351.5 |
| 9 | Finland | 1469.8 | 2961.59 | 1221.90 | 0.00 | 1531.32 | 1757.78 | 770.3 |
| 10 | France | 30,910.8 | 44,481.16 | 10,554.53 | 1533.83 | 8009.23 | 25,994.09 | 949.9 |
| 11 | Germany | 20,457.7 | 36,231.50 | 10,738.98 | 251.68 | 6487.51 | 15,590.69 | 939.5 |
| 12 | Greece | 6221.0 | 5880.03 | 1197.55 | 400.51 | 2539.28 | 7214.77 | 1841.9 |
| 13 | Hungary | 3420.4 | 5055.95 | 1085.52 | 32.76 | 1386.93 | 3742.56 | 760.4 |
| 14 | Ireland | 3276.7 | 5704.41 | 1009.48 | 38.22 | 1727.31 | 4008.48 | 814.7 |
| 15 | Italy | 32,470.8 | 24,959.42 | 10,071.44 | 604.01 | 4187.27 | 25,466.58 | 2033.4 |
| 16 | Latvia | 590.1 | 1138.40 | 142.80 | 20.41 | 317.54 | 745.14 | 378.4 |
| 17 | Lithuania | 1499.8 | 1958.05 | 391.10 | 1.11 | 246.36 | 1332.37 | 457.1 |
| 18 | Luxembourg | 126.1 | 299.70 | 95.61 | 1.77 | 71.59 | 94.62 | 716.8 |
| 19 | Malta | 53.4 | 66.69 | 7.25 | 0.00 | 29.07 | 74.41 | 7441.0 |
| 20 | Netherlands | 10,571.7 | 17,329.60 | 4306.92 | 348.52 | 1436.30 | 7152.50 | 3934.3 |
| 21 | Poland | 10,305.6 | 15,968.14 | 1767.85 | 438.26 | 4002.03 | 12,017.04 | 812.8 |
| 22 | Portugal | 3305.1 | 4998.24 | 850.62 | 49.64 | 886.29 | 3226.19 | 813.9 |
| 23 | Romania | 8273.6 | 8391.82 | 4251.80 | 24.95 | 2379.80 | 6222.99 | 487.6 |
| 24 | Slovakia | 641.8 | 1666.71 | 262.90 | 51.14 | 580.53 | 893.47 | 479.6 |
| 25 | Slovenia | 589.1 | 771.48 | 271.77 | 3.29 | 268.20 | 574.90 | 1190.3 |
| 26 | Spain | 27,841.4 | 23,655.81 | 5463.74 | 443.99 | 5634.78 | 27,231.24 | 1138.7 |
| 27 | Sweden | 1727.4 | 4313.12 | 1075.91 | 0.00 | 887.20 | 1488.27 | 495.1 |
| 28 | EU 27 (from 2020) | 178,370.6 | 232,848.51 | 63,202.23 | 4726.95 | 52,208.28 | 166,170.61 | 1055.6 |

Source: Authors' processing, according to Eurostat (2020) data.

The analysis of the structure for total intermediate consumption for EU 27 indicates a large share owned by feeding stuff, followed by energy and fertilizers (Figure 7).

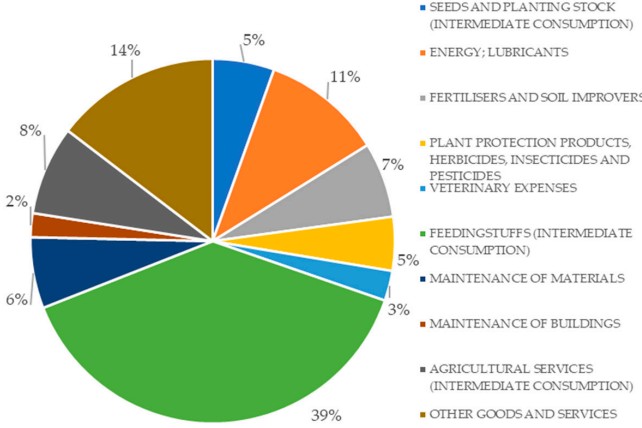

**Figure 7.** Total intermediate consumption structure at the level of the EU, 2020. Source: Authors' calculation and representation, according to Eurostat (2020) data.

The global performance in agriculture is not only given by the size of the result obtained but also by input use efficiency [70]. Within the input value structure for EU 27, the largest share is represented by feeding stuff followed by energy (Figure 8).

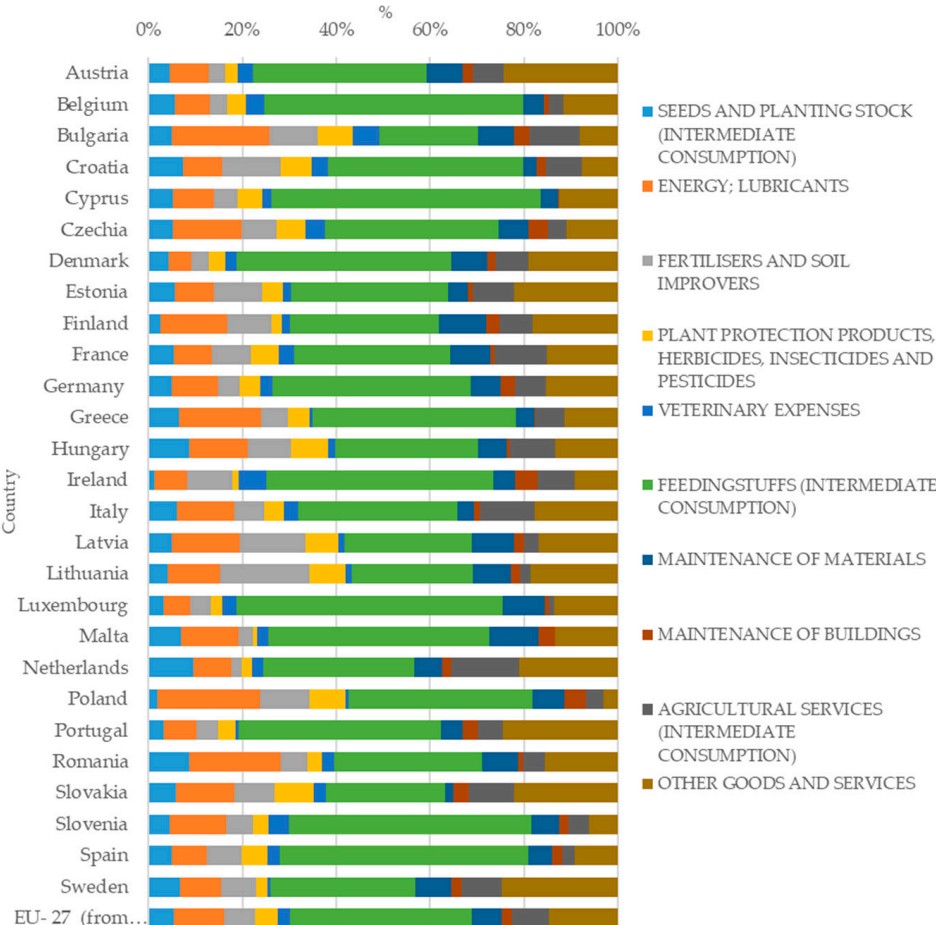

**Figure 8.** Total intermediate consumption structure for EU Member States, 2020. Source: Authors' calculation and representation, according to Eurostat (2020) data.

The main indicator referred to by the current paper and research is output/input ratio. Not the amount of output (result) but the correlation of the output with the input/consumption provides the most accurate image (overview) on agricultural performance and the economic efficiency of the activity. The total input value was calculated as total intermediate consumption + fixed capital consumption + compensation of employees + rents and other charges to be paid + interest paid − interest received.

The highest output/input ratios are recorded by Malta (1.509), Cyprus (1.451), and Spain (1.411) while the lowest values belong to Slovakia (0.889), Estonia (0.899), and Finland (0.917) (Table 11). The table invalidates the second research hypothesis, according to which the output–input ratio for the EU Member States as performance indicator is related to agricultural output and correlates with agricultural results for each country, indicating a significant difference between the value of the output for the agricultural industry and output/input ratio for the most of the countries, the hierarchy being significantly altered.

**Table 11.** Output/input ratio calculation based on economic results and consumptions (million euro).

| No. | Country/Item | Output of the Agricultural Industry | Total Intermediate Consumption | Fixed Capital Consumption | Compensation of Employees | Rents and Other Charges to be Paid | Interest Paid | Interest Received | Output /Input Ratio |
|---|---|---|---|---|---|---|---|---|---|
| 1 | Austria | 7645.59 | 6504.04 | 812.30 | 466.2 | 225.0 | 21.4 | 102.5 | 0.965 |
| 2 | Belgium | 9057.99 | 4410.47 | 1878.16 | 682.3 | 244.3 | 234.4 | 0.0 | 1.216 |
| 3 | Bulgaria | 4021.96 | 2181.84 | 453.27 | 543.6 | 595.1 | 31.2 | 11.9 | 1.060 |
| 4 | Croatia | 2422.11 | 1291.14 | 316.27 | 153.7 | 41.7 | 26.8 | 10.2 | 1.329 |
| 5 | Cyprus | 759.14 | 401.38 | 15.79 | 90.9 | 11.1 | 4.0 | 0.0 | 1.451 |
| 6 | Czechia | 5632.71 | 3652.94 | 773.94 | 1246.4 | 325.0 | 65.1 | 30.6 | 0.934 |
| 7 | Denmark | 11,617.20 | 7832.34 | 1141.14 | 1184.8 | 591.9 | 51.7 | 83.7 | 1.083 |
| 8 | Estonia | 995.34 | 742.55 | 150.57 | 157.8 | 37.1 | 21.2 | 1.3 | 0.899 |
| 9 | Finland | 4475.27 | 2961.59 | 1221.90 | 408.9 | 197.1 | 88.7 | : | 0.917 |
| 10 | France | 76,630.72 | 44,481.16 | 10,554.53 | 7602.6 | 2426.4 | 470.8 | 62.0 | 1.170 |
| 11 | Germany | 57,345.40 | 36,231.50 | 10,738.98 | 5519.4 | 3349.4 | 889.4 | 37.7 | 1.012 |
| 12 | Greece | 12,051.74 | 5880.03 | 1197.55 | 650.9 | 436.5 | 209.2 | 0.0 | 1.439 |
| 13 | Hungary | 8398.47 | 5055.95 | 1085.52 | 1288.8 | 416.5 | 10.0 | 16.2 | 1.071 |
| 14 | Ireland | 8891.82 | 5704.41 | 1009.48 | 812.0 | 477.5 | 34.5 | 0.0 | 1.106 |
| 15 | Italy | 57,833.34 | 24,959.42 | 10,071.44 | 8073.1 | 1556.7 | 1014.6 | 0.0 | 1.266 |
| 16 | Latvia | 1727.16 | 1138.40 | 142.80 | 188.7 | 43.3 | 7.9 | 1.1 | 1.136 |
| 17 | Lithuania | 3486.40 | 1958.03 | 391.10 | 409.8 | 98.6 | 9.5 | 0.5 | 1.216 |
| 18 | Luxembourg | 439.85 | 299.70 | 95.61 | 31.0 | 20.5 | 2.5 | 0.0 | 0.979 |
| 19 | Malta | 121.09 | 66.69 | 7.25 | 5.2 | 0.7 | 0.4 | 0.0 | 1.509 |
| 20 | Netherlands | 28,235.59 | 17,329.60 | 4306.92 | 3134.2 | 673.4 | 724.1 | 82.6 | 1.082 |
| 21 | Poland | 26,405.78 | 15,968.14 | 1767.85 | 1750.0 | 106.3 | 221.9 | 34.6 | 1.334 |
| 22 | Portugal | 8403.49 | 4998.24 | 850.62 | 1024.4 | 34.8 | 169.3 | 6.4 | 1.188 |
| 23 | Romania | 16,824.17 | 8391.82 | 4251.80 | 1103.0 | 140.1 | 125.1 | 6.2 | 1.201 |
| 24 | Slovakia | 2348.02 | 1666.71 | 262.90 | 590.8 | 85.6 | 44.0 | 7.8 | 0.889 |
| 25 | Slovenia | 1370.29 | 771.48 | 271.77 | 76.8 | 21.0 | 2.7 | 0.6 | 1.199 |
| 26 | Spain | 51,787.23 | 23,655.81 | 5463.74 | 5842.7 | 1377.2 | 364.7 | : | 1.411 |
| 27 | Sweden | 6170.33 | 4313.12 | 1075.91 | 330.0 | 301.6 | 167.8 | 46.0 | 1.005 |
| 28 | EU 27 (from 2020) | 415,098.18 | 232,848.51 | 63,202.23 | 43,368.1 | 13,834.3 | 5012.7 | 541.9 | 1.160 |

Source: Authors' calculation, according to Eurostat (2020) data; :–not available.

Starting from the calculations related to the previous table, the situation of each country was graphically represented in relation to the EU average regarding output/input ratio. The vertical bars are associated with the level of each country and the dotted line with the EU average level. It is observed that countries such as Slovakia, Estonia, Finland, Czechia, or Austria have a ratio below the EU average (1.16) while countries such as Malta, Cyprus, Greece, or Spain are the top-leaders in output/input ratio within EU 27 (Figure 9).

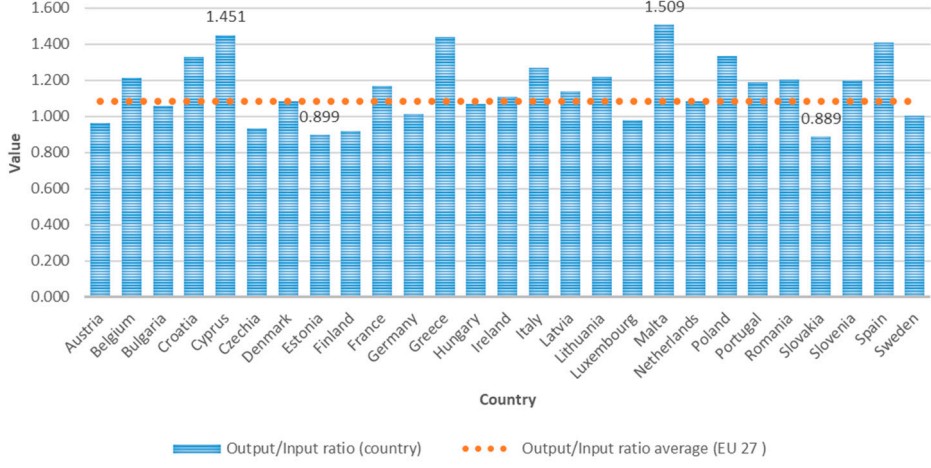

**Figure 9.** Output/input ratio for EU Member States in relation to the EU average, 2020. Source: Authors' calculation and representation, according to Eurostat (2020) data.

Based on three indicators (output/input ratio, gross fixed capital formation per UAA, and factor income), the hierarchical cluster analysis was used with distance matrix based on Euclidean coefficient, creating a dendrogram plot which indicates groups of countries with close similarities (Figure 10). The figure validates the third research hypothesis indicating that within the CAP strategy, the EU agricultural development among the EU Member

States registers a common foundation/base as concentration degree, as the countries are close to the EU mean with acceptable variations.

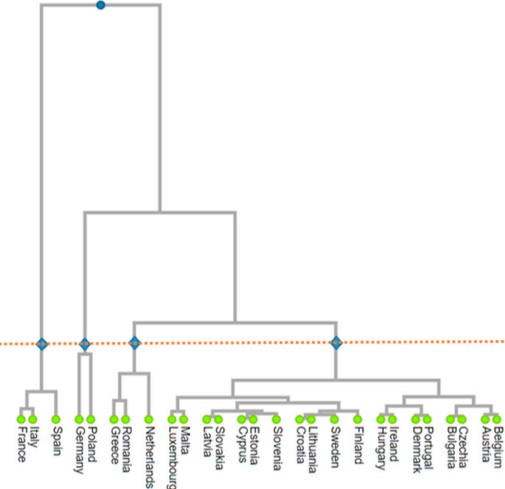

**Figure 10.** The dendrogram associated to hierarchical cluster analysis. Source: Authors' calculation and representation, according to Eurostat (2020) data.

There are four fundamental groups at the level three of segmentation. The cophenetic correlation coefficient (CP) = 0.966135201280996, which shows an efficient clustering based on the chosen variables.

The dendrogram divides the analyzed countries into four classes (at the fourth level of analysis, identified by the orange line, using the bottom up view), highlighting the degree of similarity between them based on the three indicators taken into account. Class 1 consists of Spain, France, and Italy, class 2 is formed by Poland and Germany, class 3 being represented by the Netherlands, Romania, and Greece, and class 4 containing the rest of the member countries (Figure 11).

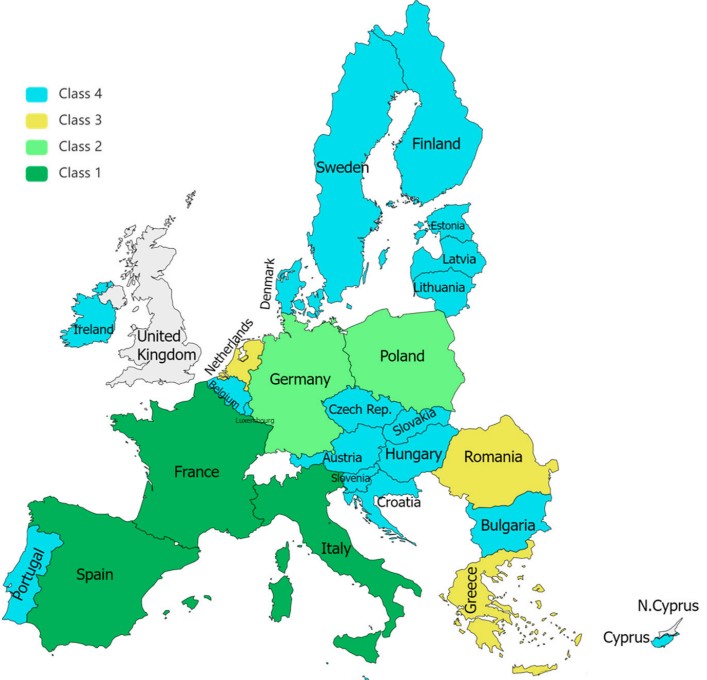

**Figure 11.** Graphic representation of the distribution for the clusters generated by hierarchical cluster analysis. Source: Authors' calculation and representation, according to Eurostat (2020) data.

This grouping, according to the three mentioned indicators, indicates a new way to approach the issue of agricultural performance, the ranking of countries in this sector, and the criteria of similarity between them.

## 5. Conclusions

Starting from the limits of previous research, the paper analyzed the performance of agriculture in the European Union from the perspective of specific factors and determinants in the year 2020. The current paper included a series of comparisons and analyses that allowed the identification of some necessary correlations for shaping an overall vision, validating the hypotheses related to the direct positive influence of gross fixed capital formation on total agricultural output and the common foundation/base as concentration degree among the EU Member States agriculture and invalidating the hypothesis related to the direct relationship of output–input ratio with agricultural output/result.

A thorough investigation of the aspects regarding the efficiency of inputs, the level of gross fixed capital formation and the output–input ratio revealed a change in the hierarchy regarding the performance of agriculture compared to the classic hierarchy, based on agricultural output, and highlighted a grouping of countries (with the help of the dendrogram) which requires increased attention in future analyses of EU agriculture.

The timeliness of the study is reflected by the importance of agriculture as a strategic sector in development and ensuring food security, and by the need to evaluate the level of development of each country as accurately as possible in order to outline European agricultural policies, as well as by identifying new relationships between management indicators of agricultural resources.

The limitations of the scientific work are given by the focus on a research direction regarding the level analysis of agricultural performance and not the incremental analysis/dynamics and the lack of a specific analysis of the specific elements of each country (climate, national agricultural policies, agricultural land area) through their integration in a statistical–mathematical model but they can represent the starting point of some subsequent scientific works.

The future directions of research, starting from the obtained results could also be represented by the in-depth analysis of output/input ratio groups by economic size, by development regions or by the training level of farmers according to age or gender or the study of potential correlations between the use of inputs, gross fixed capital formation, agricultural output or factor income. Moreover, the research team is considering the development of an algorithm that takes into account the cumulative influence of the rainfall (as a fundamental climate factor), the soil and the inputs used on the results obtained, and in general on the agricultural performance of each country in the European Union, starting from specific localized case studies in order to highlight as faithfully as possible the influence of the each country specifics/limitations on the level of agricultural development. At the same time, another targeted research direction is represented by the incremental analysis (2016–2020) of the agricultural performance level in order to highlight the dynamics of specific indicators and the influence of the accumulation/increase in technical equipment (as gross fixed capital formation) on the level of performance in the years following the investments made.

The study presents replicability both temporally (it can be repeated and compared) and spatially (it can be adapted to the level of each country or region).

**Author Contributions:** Conceptualization, G.Ș.; methodology, Ș.V.; software, Ș.V.; validation, G.Ș., O.C., I.S.B. and D.C.; formal analysis, G.Ș. and Ș.V.; investigation, G.Ș., Ș.V., O.C., I.S.B. and D.C.; resources, Ș.V., O.C. and D.C.; data curation, G.Ș. and I.S.B.; writing—original draft preparation, Ș.V. and D.C.; writing—review and editing, Ș.V. and O.C.; visualization, D.C. and Ș.V.; supervision, D.C. and G.Ș.; project administration, O.C. All authors have read and agreed to the published version of the manuscript.

**Funding:** This research was funded by a grant from European Regional Development Fund through the Competitiveness Operational Program 2014–2020, project AGRIECOTEC-SMIS code 119611, ID: P_40_385_CF_POC123G_2018 (contract no. 4/AXA1/1.2.3G/05.06.2018).

**Institutional Review Board Statement:** Not applicable.

**Informed Consent Statement:** Not applicable.

**Data Availability Statement:** Not applicable.

**Acknowledgments:** This research is part of the project "Establishment and implementation of partnerships for the transfer of knowledge between the Iasi Research Institute for Agriculture and Environment and the agricultural business environment", acronym "AGRIECOTEC".

**Conflicts of Interest:** The authors declare no conflict of interest.

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
