# Peer review of "Analysis of the Determinants of Agriculture Performance at the European Union Level"

_agriculture, doi:10.3390/agriculture13030616_

Round 1

Reviewer 1 Report

Main Comments

The main concern is the methodology adopted, and thus the results might need to be interpreted with caution. Correlation between two variables does not imply causality.

1.      The acronym EU in the title should first be defined in full before writing the abbreviation.

2.      I suggest changing the title to ‘Analysis of the determinants of Agriculture Performance at the European Union Level’.

3.      Figure 9 on page 20, is not clear and not well explained. How can a layman read and understand this figure?

Minor Comments

1.      On page 1, under the introduction, lines 36-38. This statement needs to be reframed ‘ The performance of agriculture is a result of several determinants and characteristic factors with conditioning, complementarity or competitiveness relationships established between…..

2.      All the acronyms such as CAP (Common Agricultural Policy) should be defined where it was first used e.g. page 2, line 89. Similarly, ‘Annual work units (AWUs)’, etc.

3.      All the figures should be put in the appendix.

4.      The title of Table 11. Output/Input ratio calculation based on results and consumption values (million euro euro) should be checked.

5.      Check the manuscript for minor typo errors such as on page 20, line 473 for these minor typo errors ‘ The Eu average regarding Output/Input ratio is’, Czechia

Reviewer 2 Report

The article is very well prepared, but in my opinion it has quite serious shortcomings, i.e.:

(a) countries with different characteristics of agriculture in different climatic zones are compared, and therefore the limits of the productivity of these countries are different;

(b) the development of agriculture varies.

In my opinion, the work would be more valuable if:

(a) the maximum possibilities of agriculture, with today's technology, and the degree of use of these possibilities in the countries concerned were determined;

or alternatively

(b) the level of technical equipment and efficiency in previous years was determined and it was checked how the increase in technical equipment affects changes in efficiency and changes in production concentration (i.e. the study should concern increments, not levels)
